# Sensor orientation and other factors which increase the blast overpressure reporting errors

Anthony Misistia[1,2☯], Maciej Skotak[1,3☯]*, Arturo Cardenas[1,2], Eren Alay[3,4], Namas Chandra[3], Gary H. Kamimori[1]*

**1** Blast Induced Neurotrauma Department, Walter Reed Army Institute of Research, Silver Spring, MD, United States of America, **2** Cherokee Nation Assurance, Catoosa, OK, United States of America, **3** Department of Biomedical Engineering, New Jersey Institute of Technology, University Heights, Newark, NJ, United States of America, **4** Department of Computer Science, Stevens Institute of Technology, Hoboken, NJ, United States of America

☯ These authors contributed equally to this work.

* maciej.skotak.ctr@mail.mil (MS); gary.h.kamimori.civ@mail.mil (GHK)

## Abstract

This study compared the response of the wearable sensors tested against the industry-standard pressure transducers at blast overpressure (BOP) levels typically experienced in training. We systematically evaluated the effects of the sensor orientation with respect to the direction of the incident shock wave and demonstrated how the averaging methods affect the reported pressure values. The evaluated methods included averaging peak overpressure and impulse of all four sensors mounted on a helmet, taking the average of the three sensors, or isolating the incident pressure equivalent using two sensors. The experimental procedures were conducted in controlled laboratory conditions using the shock tube, and some of the findings were verified in field conditions with live fire charges during explosive breaching training. We used four different orientations (0˚, 90˚, 180˚, and 270˚) of the headform retrofitted with commonly fielded helmets (ACH, ECH, Ops-Core) with four B3 Blast Gauge sensors. We determined that averaging the peak overpressure values overestimates the actual dosage experienced by operators, which is caused by the reflected pressure contribution. This conclusion is valid despite the identified limitation of the B3 gauges that consistently underreport the peak reflected overpressure, compared to the industry-standard sensors. We also noted consistent overestimation of the impulse. These findings demonstrate that extreme caution should be exercised when interpreting occupational blast exposure results without knowing the orientation of the sensors. Pure numerical values without the geometrical, training-regime specific information such as the position of the sensors, the distance and orientation of the trainee to the source of the blast wave, and weapon system used will inevitably lead to erroneous estimation of the individual and cumulative blast overpressure (BOP) dosages. Considering that the 4 psi (~28 kPa) incident BOP is currently accepted as the threshold exposure safety value, a misinterpretation of exposure level may lead to an inaccurate estimation of BOP at the minimum standoff distance (MSD), or exclusion criteria.

**Data Availability Statement:** Data are available as an Excel file in the supporting information (PONE-D-20-17170 - Experimental data.xlsx)

**Funding:** This work was supported by U.S. Army MRMC MOMRP (RAD 3) grant "Environmental Sensors in Training" (ESiT), and Office of the Assistant Secretary of Defense for Health Affairs, award no. W81XWH-16-2-0001 (PI: Dr. Gary H. Kamimori). The funders had no role in study design, data collection and analysis, decision to publish, or preparation of the manuscript.

**Competing interests:** The authors have declared that no competing interests exist.

## Introduction

Military personnel are often exposed to blast overpressure during training with explosives as well as heavy (e.g., mortars, rockets & machine guns) and light (rifles & pistols) weapons and during combat [1, 2]. Recent research has demonstrated that exposure to blast overpressure can result in deficits in neurocognitive performance [3, 4] and changes in blood biomarkers [5–9] in the absence of a medically diagnosable injury such as concussion. At present any acute or subacute reported symptoms short of a medically diagnosed injury are not entered into an individual's medical record or other service records. In response to this emerging information, the recently enacted FY2020 National Defense Authorization Act (NDAA) stipulates, in section 717, the requirement for monitoring of blast exposure in training and combat, and its inclusion in the service member's medical record [10]. According to that Public Law (116–92), the blast exposure history should include, at a minimum, the date of exposure, duration of exposure, and blast pressure. There is currently no guidance on how these measurements will be implemented, including logistics associated with data processing and storage.

In conjunction with the development and fielding of the Black Box Biometrics (B3) Blast Gauge® sensors, DARPA developed a three-sensor configuration [11], which can be considered as the starting point of the development of the standard for a blast overpressure monitoring program in military training and operations. For that reason, the B3 sensors are imprinted with a labeling scheme identifying a specific location for the sensor to be worn: back of the helmet (H), on the left shoulder (S), and chest (C). Similarly, at present no promulgated guidelines or doctrine exists regarding whether a single sensor data or an average of two, three or more sensors should be used to estimate an individual's occupational blast exposure. Nor are there codified acquisition requirements for other blast parameters recorded by wearable sensors, or conditions of program implementation, such as sensor sampling rate or prescribed use scenarios. It is essential to recognize that understanding an exposure to a blast wave using a limited number of sensors is inherently flawed, considering the complexity of a given blast wave propagation through a specific environment. But it is also an approach that is dictated by the reality of military operations where high-fidelity blast exposure monitoring is not the highest priority.

The primary instrumental factor a pressure transducer should have is a fast dynamic performance [12], and sensor construction must account for the harsh operational environment. More importantly, the three chief non-instrumental factors that might influence the recorded pressure values from wearable sensors are: 1) the orientation of the body with respect to the source(s) of the blast wave(s), 2) the intensity of the incident shock wave, and 3) the local geometry. For a simple case of a pressure transducer mounted flush with a surface of a flat plate, at a zero-degree incident angle (sensing element facing the blast), the reflected pressure will be recorded, whereas at 90-degrees (sensing element parallel to the blast direction), a pure incident shock wave will be registered. The transition (0 to 90 degrees) incident angles result in a 'mixed-mode' pressure waveform, with gradually decreasing contribution of the reflected pressure. The relationship between the incident and reflected shock waves is not intuitive, and it has been extensively researched to understand the mechanistic aspects of shock wave reflections from flat surfaces [13–16]. The loading caused by blast waves on buildings is also an area of great interest in civil engineering where reflected pressure is used to evaluate the structural integrity [17–20].

The instrumental factors relevant to the pressure wave quantification are described and discussed in detail. However, in this paper, the descriptor that is most important for disputing the accuracy of dosage evaluation is the sensor orientation, and a relative contribution of the incident versus the reflected pressures seen in the shock tube and field experiments is discussed. For accurate determination of the pressure waveform, especially the peak values, an immediate response of the sensing element is required. Finally, findings from the shock tube experiments were validated during the heavy wall breaching exercise.

## Materials and methods

### Instrumentation

The evaluation of the sensor performance and the effects of the helmet orientation on the pressure distribution was done in the shock tube. The 28" x 28" square cross-section shock tube was extensively validated to reproduce high-explosive blast waves in a wide range of intensities, and its operating principles are reported in several recently published references [21–24]. All tests described in this work used compressed helium to represent an idealized version of the blast wave i.e., the one that would possess controlled and repeatable characteristics [25, 26]. The test section has an array of sensor ports along the top of the tube's wall to capture the overpressures of the propagating incident shock wave impacting the test subject. The incident pressure is measured using Tourmaline ICP® pressure sensors model 134A24 (PCB Piezotronics, Depew, NY). We also used ICP® high-frequency pressure sensors model 102B06 (PCB Piezotronics, Depew, NY) as a reference for reflected pressure measurements (see next section). The sampling rate for PCB sensors was 1.0 MHz for all experiments.

The sensors used in both the field and laboratory tests were Black Box Biometrics (B3) Blast Gauge® (generation 6) sensors with overpressure ranges from 2.5 to 100 psi [11]. These sensors were always programmed ahead of time and turned on close to the time of experimentation to preserve as much battery life as possible. The sampling rate of the B3 sensors is 200 kHz, and the integration time is 20 ms.

### Incident and reflected overpressure measurement fidelity

A series of tests were conducted in a lab using a shock tube to compare the response of the B3 Blast Gauge® against two types of piezoelectric sensors made by PCB Piezotronics (Depew, NY) (models 134A24 and 102B06). The model 134A24 was mounted flush with the shock tube wall and was used as a reference to recording the incident overpressure (denoted as T5, see Fig 1 in ref. [21] for details), while three sensors were attached to the 6 mm thick rectangular aluminum blocks matching the front cross-sectional area of the B3 Blast Gauge. Three PCB sensors (model 102B06) were then mounted on the test fixture (6 x 6 inches) used as a reference to measure the reflected overpressure (Fig 1A and 1B). Next, the B3 Blast Gauges were attached using the elastic mounting band supplied with the sensor. A total of 23 sensors were used to measure the incident overpressure, while 16 sensors were used to evaluate the response in the reflected pressure mode. The incident pressure sensors were mounted at the 90˚ incident angle, while the reflected pressure sensors were mounted at the 0˚ incident angle, with respect to the direction of the shockwave propagation (facing the incoming shock wave, Fig 1B). The sensor arrays were exposed to shockwaves with a nominal peak overpressure of 5 and 10 psi (approx. 35 and 70 kPa, respectively), and each condition was repeated 10 times. The peak overpressure and impulse values of the B3 sensors were generated by the software embedded in the sensors, tabulated, and averaged (Fig 2). The reflection coefficients for peak overpressure and impulse were calculated using data

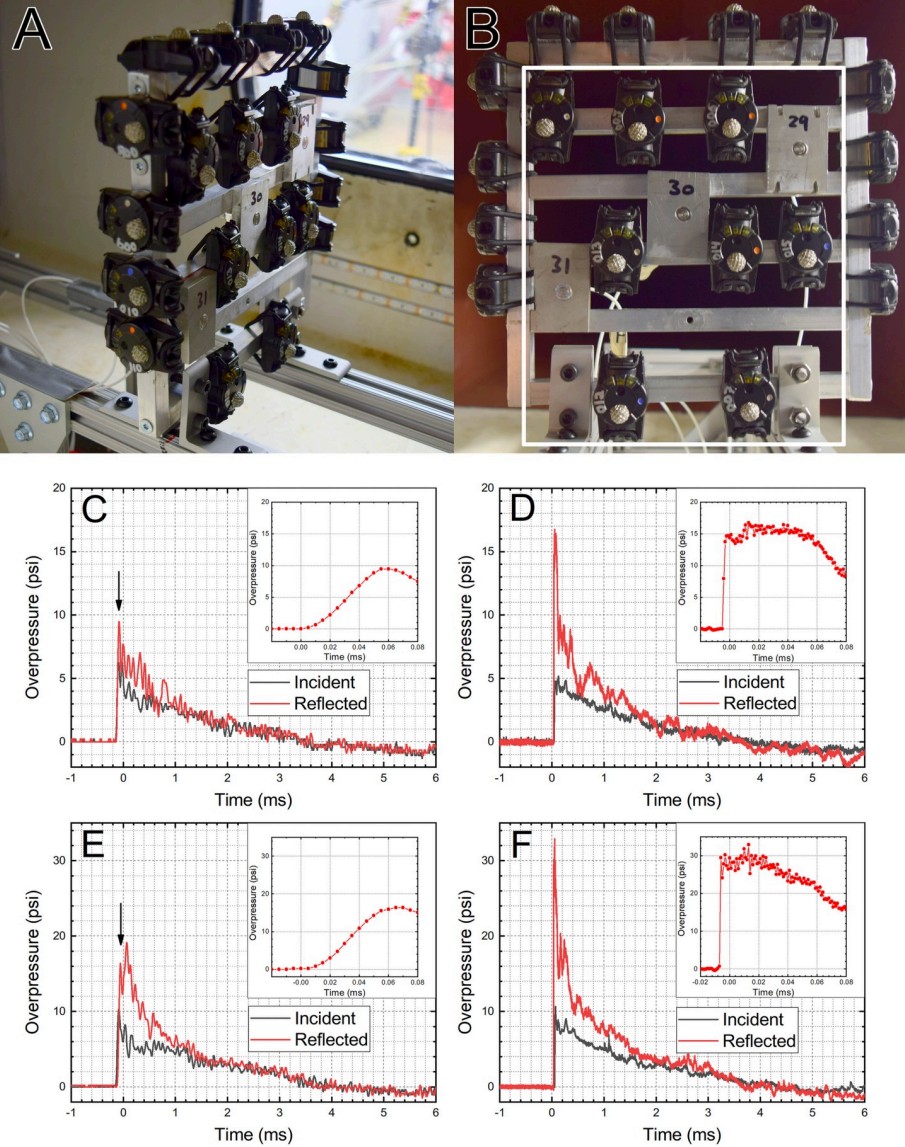

**Fig 1. The experimental setup for the shock tube experiments.** The 12 B3 sensors mounted around the perimeter are in the incident pressure orientation (A), while 8 B3 and 3 PCB sensors inside of the white rectangle are in the reflected pressure orientation and reflected pressure measurement configuration (B). Two nominal shock wave intensities were used: 5 psi (C, D) and 10 psi (D, F). Representative pressure waveforms for B3 (C, E) and PCB (E, F) sensors are presented. Insets: the reflected overpressure reported by B3 and PCB sensors in the initial 0.08 milliseconds, reveals striking differences in the dynamic response.

obtained from sensors grouped by the manufacturer [21]. The data recorded by PCB sensors model 102B06 and 134A24 were used as reflected and incident overpressure, respectively, while for the B3 sensors, the reflection coefficients (Eq 1) were calculated using sensors positioned to measure reflected and incident overpressure.

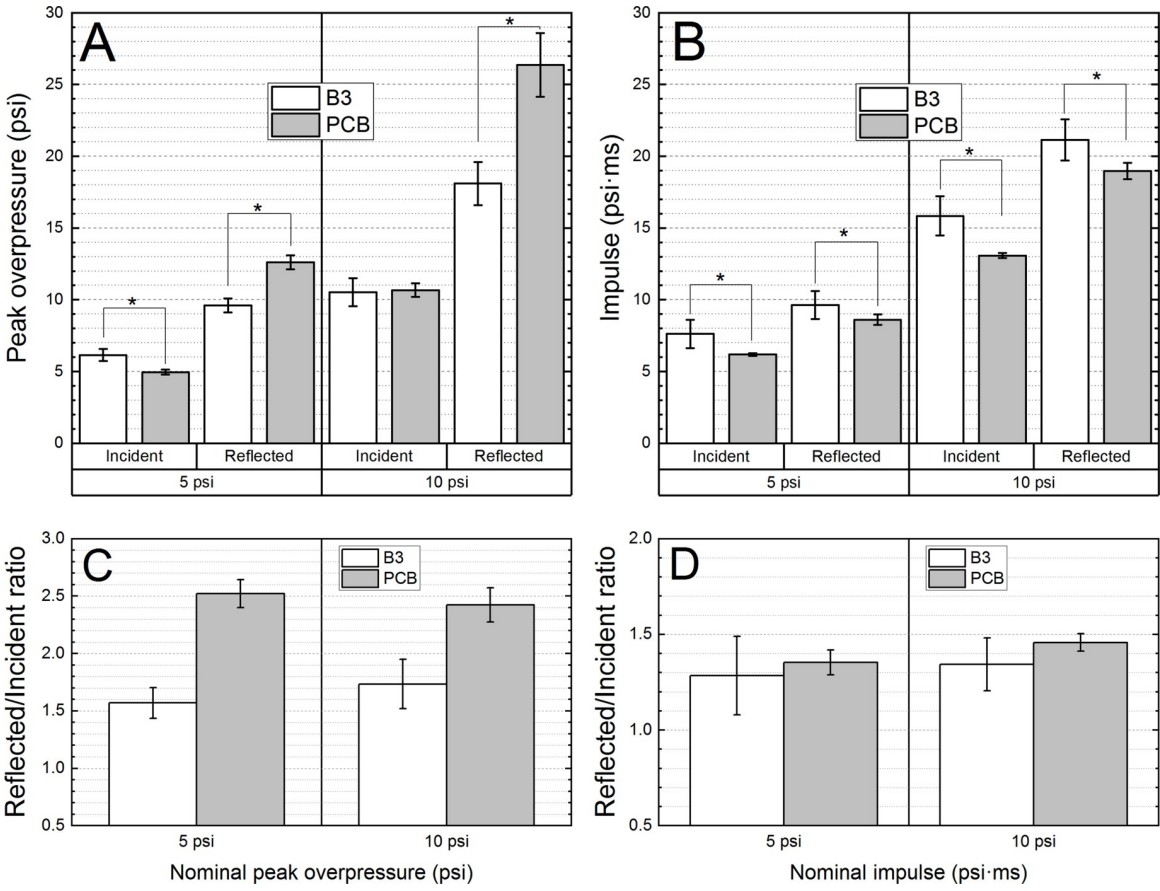

**Fig 2. The results of the quantification of the incident and reflected pressure measurements for the B3 Blast Gauge sensors with the PCB sensors used as a reference.** The average peak overpressure (A) and impulse (B) and reflected-to-incident peak overpressure (C) and impulse (D) ratios were compared. The statistical significance is marked with an asterisk (p < 0.001).

### Orientation tests

In this battery of tests, we used the human head surrogate wearing a standard-issue Advanced Combat Helmet (ACH). The helmet was instrumented with four B3 Blast Gauges (generation 6) on a band encircling the helmet's rim. The sensors were located at the front, left, right, and back of the helmet (Fig 3). The headform assembly was mounted to the mounting plate in the test section of the tube at 0˚, 90˚, 180˚, and 270˚ and exposed to a single shockwave with nominal BOP of 5 or 10 psi (approx. 35 and 70 kPa, respectively). The tests were repeated six times at each combination of headform orientation and shockwave intensity (n = 6). The peak overpressure and impulse values were quantified and plotted (Figs 3 and 4).

### Field data

Human head surrogates were mounted five feet from the ground atop range poles. The heads were outfitted with Ops-Core and ECH helmets that were instrumented with a band of four B3 Blast Gauge® generation 6 sensors (front, left, right, back) circling the helmet, like the shock tube tests. The poles and helmets were then placed at 12 m from the charges, to mirror the stack position normally taken by human subjects. The 12 m is 1.5 MSD estimated using the k-equation [2]. A single charge strength of 4.74 kg N.E.W. TNT (net explosive weight in TNT equivalency) was used in these experiments, and the exposure was repeated six times.

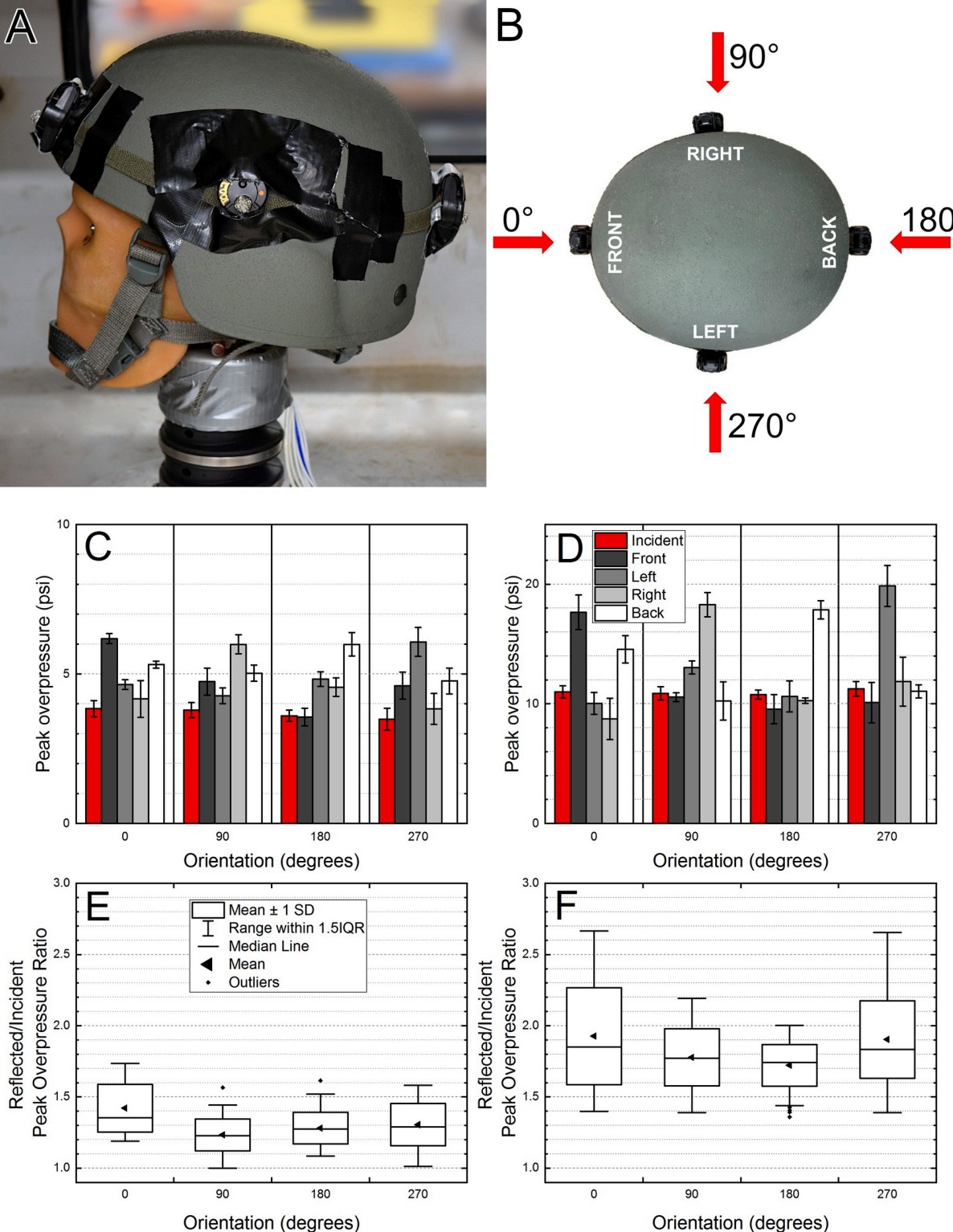

**Fig 3.** The quantification of the overpressure waveforms recorded by the B3 Blast Gauge sensors attached on the ACH helmet (A). Four sensors were used to measure the external pressure distribution on the helmet, which was rotated in 90 degrees intervals (B), and the direction of shock wave propagation is indicated by red arrows. The average overpressure reported by individual sensors exposed to a single shock wave with nominal intensity of 5 psi (C) or 10 psi (D) is presented. The reflected-to-incident peak overpressure (E) and impulse (F) ratios are also compared.

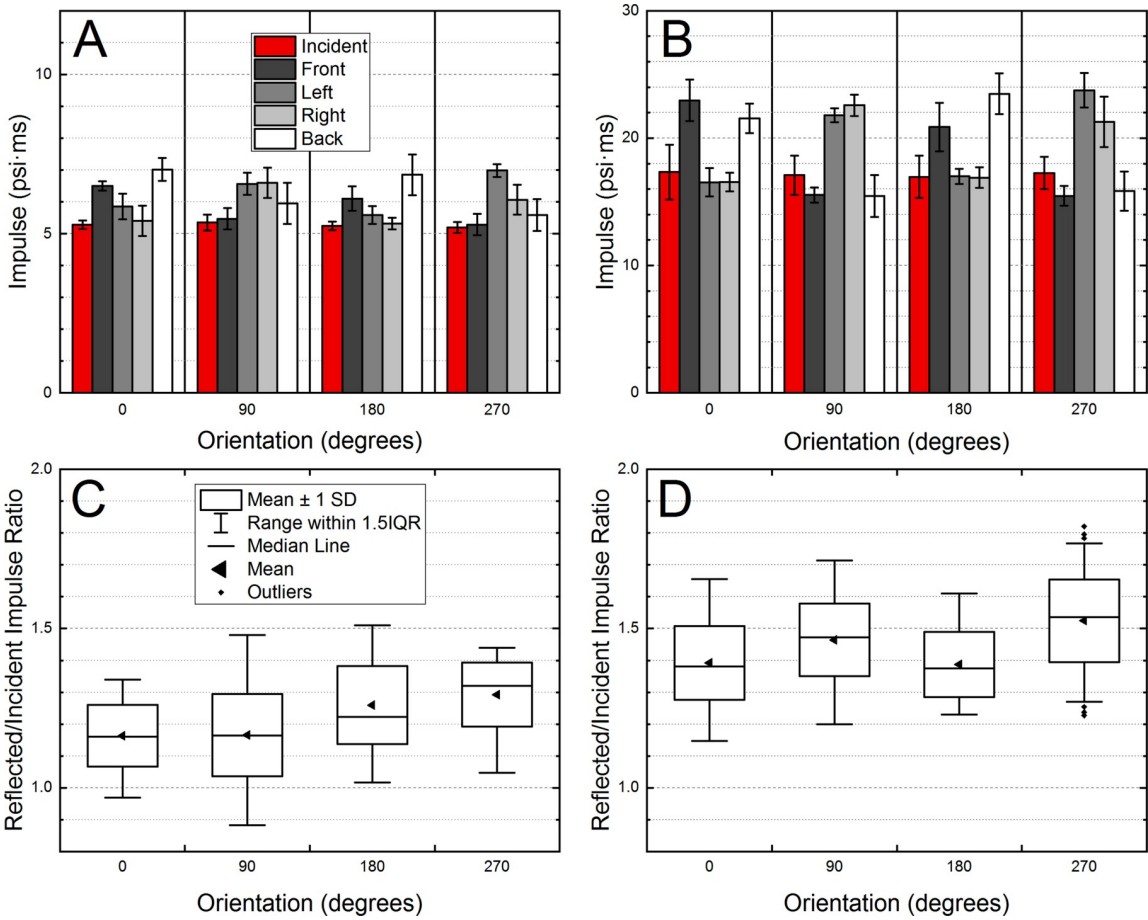

**Fig 4. The results of the quantification of the overpressure waveforms recorded by the B3 Blast Gauge on the ACH helmet.** Four sensors were used, and the helmet was rotated in 90 degrees intervals. The impulse at 5 psi (A) and 10 psi (B), and corresponding reflected-to-incident ratios (C, D) were compared.

### Reflection coefficients

Reflection coefficients were calculated using Eq 1:

$$\Lambda_n = \frac{P_r}{P_i} \qquad (1)$$

Where, the $P_r$ and $P_i$ are the reflected and incident shock overpressure, respectively [21].

In the pressure fidelity recording tests the reflection coefficients for PCB sensors were calculated using an average of 30 coefficients per BOP: 3 reflected overpressure sensors, 1 incident pressure sensor, measurements repeated 10 times per BOP. The reflection coefficients for B3 sensors used a combination of incident (12 and 11 sensors) and reflected pressure (2 x 8 sensors), which were repeated 10 times at both BOPs. The total number of reflection coefficients in these calculations is thus 960 and 880, for 5 and 10 psi, respectively.

The reflection coefficients for the ACH helmet were calculated using a permutation of the reflected pressure sensor (n = 6) and both incident pressure sensors (n = 6 for each sensor). The total number of amplification factors at a specific helmet orientation is thus 72.

### Data processing and statistical evaluation

All waveforms recorded by PCB sensors were imported, processed, and quantified in the Origin 2018 software (OriginLab Corp., Northampton, MA). Data from experiments performed at different experimental conditions (shock wave intensity, headform orientation) were pooled together into respective subsets according to blast intensity (5 or 10 psi BOP).

We also performed an analysis of the peak overpressure and impulse averaging. We used tabulated B3 sensor data for the same shot number (blast exposure); the three and four sensor averages were calculated. The effects of averaging different sensor combinations are evaluated using statistical and graphic methods (Figs 5 and 6).

The multiple comparison, independent sample two-tailed t-test, was performed with Bonferroni correction, and p < 0.001 was considered statistically significant. All data are presented as mean and standard deviation.

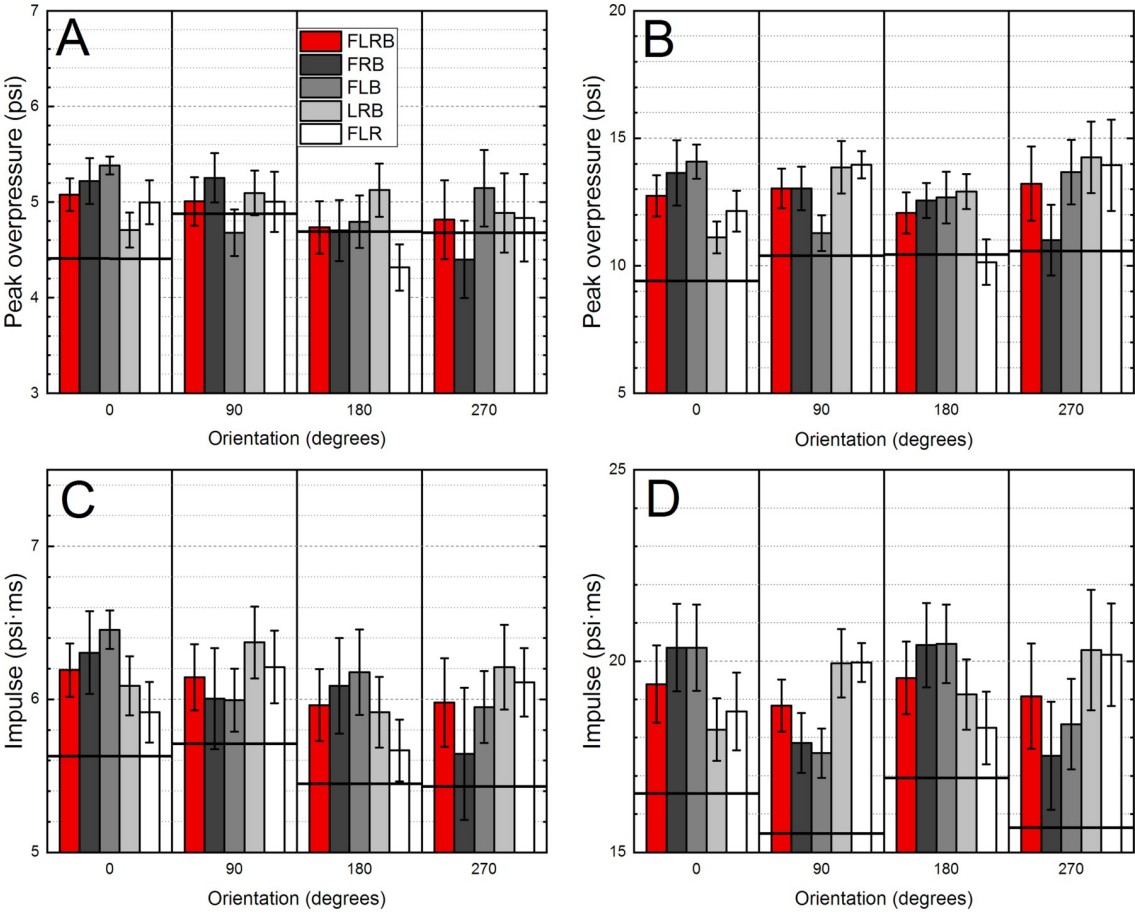

**Fig 5.** The results of the quantification of the averaging of the 3 sensor readings, the peak overpressure (A, B) or impulse (C, D) recorded by the B3 Blast Gauge mounted on the ACH helmet. The average of four sensors (labelled as FRLB) was compared against various combinations of 3 sensor averages (labelled as: FRB, FLB, LRB and FLR). The results as a function of helmet orientation for the data collected at 5 psi (A, C) and 10 psi (B, D) are presented. The horizontal black line indicates the incident peak overpressure (or impulse) calculated as an average of two sensors: LEFT and RIGHT for 0° and 180°, or FRONT and BACK for 90° and 270° ACH orientations, respectively.

**A**

|  | 0° | | 90° | |
|  | FRB | FLB | FRB | FLB |
|---|---|---|---|---|
| 0° LRB | 2E-03 | 7E-05 | 0.29 | 0.01 |
| 0° FLR | 0.13 | 8E-03 | 0.16 | 0.08 |
| 270° LRB | 0.07 | 0.30 | 0.03 | 0.06 |
| 270° FLR | 0.11 | 0.24 | 0.04 | 0.01 |
|  | FRB | FLB | FRB | FLB |
|  | 270° | | 180° | |

(right-side labels: 90°, 180°)

**B**

|  | 0° | | 90° | |
|  | FRB | FLB | FRB | FLB |
|---|---|---|---|---|
| 0° LRB | 0.14 | 4E-03 | 0.05 | 0.01 |
| 0° FLR | 0.02 | 4E-04 | 0.25 | 0.12 |
| 270° LRB | 0.03 | 0.11 | 0.31 | 0.11 |
| 270° FLR | 0.05 | 0.25 | 0.02 | 0.01 |
|  | FRB | FLB | FRB | FLB |
|  | 270° | | 180° | |

(right-side labels: 90°, 180°)

**C**

|  | 0° | | 90° | |
|  | FRB | FLB | FRB | FLB |
|---|---|---|---|---|
| 0° LRB | 3E-03 | 1E-05 | 0.16 | 7E-04 |
| 0° FLR | 0.04 | 1E-03 | 0.05 | 3E-05 |
| 270° LRB | 2E-03 | 0.47 | 0.40 | 0.65 |
| 270° FLR | 0.01 | 0.77 | 5E-04 | 1E-03 |
|  | FRB | FLB | FRB | FLB |
|  | 270° | | 180° | |

(right-side labels: 90°, 180°)

**D**

|  | 0° | | 90° | |
|  | FRB | FLB | FRB | FLB |
|---|---|---|---|---|
| 0° LRB | 5E-03 | 4E-03 | 2E-03 | 5E-04 |
| 0° FLR | 0.02 | 2E-02 | 4E-04 | 5E-05 |
| 270° LRB | 1E-02 | 0.04 | 0.05 | 0.04 |
| 270° FLR | 0.01 | 0.03 | 5E-03 | 3E-03 |
|  | FRB | FLB | FRB | FLB |
|  | 270° | | 180° | |

(right-side labels: 90°, 180°)

**Fig 6. The heat map illustrating the statistical significance thresholds for the comparison of the averages of three sensors.** The comparison of the groups where FRONT or BACK were excluded (LRB or FLR, respectively) against groups where LEFT or RIGHT sensors were excluded (denoted as FRB or FLB, respectively) from the calculations. The averages of the peak overpressure (A, C) and impulse (B, D) of the waveforms recorded at 5 psi (A, B) or 10 psi (C, D) nominal BOP are presented. The multiple pairwise comparison was performed only at specific headform-helmet orientation: 0˚ (upper left quadrants), 90˚ (upper right quadrants), 180˚ (lower right quadrants), and 270˚ (lower left quadrants). The following p-value thresholds were used: p < 0.001 (green), 0.05 < p <0.001 (yellow), p > 0.05 (red) to construct the heat map.

## Results

### Incident and reflected overpressure measurement in the shock tube

The test rig with attached B3 Blast Gauges and PCB sensors was mounted in the shock tube on a frame made to keep them separated from the shock tube walls (Fig 1A and 1B) and, in this way, to avoid any flow obstructions, reflections, and other artifacts [25]. The reference incident overpressure was measured above the test rig by a PCB sensor, similarly, as it was done in the

past [1, 21, 25, 27], while reference reflected pressure was measured by 3 PCB sensors (marked as 29, 30 and 31 in Fig 1B). The representative pressure profiles captured in the incident and reflected pressure configurations by sensors from both manufacturers are presented in Fig 1C–1F. Additionally, the insets present the 0.1 milliseconds of the signal when the shock wave pressure increase is recorded.

The quantification of the peak overpressure and impulse and the reflected-to-incident pressure ratios is presented in Fig 2. There are some easily identifiable trends in the data: reflected peak overpressure is underreported by B3 sensors (Fig 2A), while the impulse values are overestimated (Fig 2B). For B3 sensors the incident peak overpressure is higher at 5 psi and matches the PCB sensors at 10 psi. Consequently, the reflection coefficients for the peak overpressure are largely underestimated for B3 sensors (Fig 2C). However, the reflection coefficients for the impulse are in good agreement between the two sensor types, B3 and PCB (Fig 2D).

## Effect of helmet orientation on overpressure: Laboratory and field data

The standard-issue ACH helmet was instrumented with four B3 Blast Gauges (Fig 3A), spaced at angular coordinates of 0°, 90°, 180°, and 270°, and denoted as FRONT, RIGHT, BACK and LEFT, respectively (Fig 3B). The helmet was mounted on an anthropometric headform and exposed to the shock wave with two nominal intensities of 5 and 10 psi (n = 6 for each BOP). and the helmet-headform assembly was then rotated counterclockwise by 90° intervals (90°, 180° and 270° position with respect to the original orientation, Fig 3B), and the same set of exposures was repeated. The average peak overpressure (Fig 3C and 3D) and impulse (Fig 4A and 4B) values are presented with corresponding reflection coefficients (Fig 3E and 3F for peak overpressure, and Fig 4C and 4D for impulse). There are consistent trends in the individual sensor data: 1) the sensors facing the blast wave (e.g., FRONT for the 0° orientation, Fig 3B) and these on the opposite side of the helmet (e.g., BACK for the 0° orientation, Fig 3B) have the highest peak overpressure and impulse values, 2) the sensors positioned to record the incident pressure have the lowest peak overpressure and impulse (e.g., sensors LEFT and RIGHT at 0° orientation, Fig 3B). The reflection coefficients are in general higher at higher BOPs: 1.0 to 1.72 (Fig 3E) and 0.9 to 1.5 (Fig 4C) at 5 psi, and 1.4 to 2.7 (Fig 3F) and 1.15 to 1.75 (Fig 4D) at 10 psi.

## The analysis of the overpressure averages

The results of the averaging for the three- and four-sensor combinations of the peak overpressure and impulse are presented in Fig 5. The letters in the labels indicate which sensors were used to calculate the average, e.g., the "FRONT, LEFT and BACK" configuration is marked as FLB. Similarly, the "LEFT, RIGHT, and BACK" configuration is marked as LRB, and so on. The incident pressure (indicated as a thick horizontal line) was calculated using the two sensors, which were positioned to measure this kind of a pressure waveform (Fig 5A–5D). In general, with only two exceptions (FLB at 90°, and FLR at 180°, Fig 5A), the incident peak overpressure calculated in this way is lower than any combination of three- or four-sensor averages calculated for specific helmet orientation. In the case of impulse values, the two-sensor incident impulse values are the lowest across all combinations compared (Fig 5C and 5D).

In order to gain a better understanding of the effects associated with the elimination of one of the sensors from the averaging, we created a heatmap complementary to Figs 4 and 5. The heatmap illustrates the statistical significance thresholds for multiple comparison t-test for three-sensor configurations only (Fig 6). Irrespectively on the helmet orientation, there are always two of the four sensors oriented in a position to capture incident pressure. For example,

in the LRB-FLR group, the geometry of the sensors LEFT and RIGHT dictated that they can measure incident pressure waveform for the 0° or 180° helmet orientations. Based on this criterion, the two classes are identified: 1) LRB-FLR, and 2) FRB-FLB. The total number of comparisons per group was six, and we used three thresholds for the heatmap: 1) p < 0.01 (green), 2) 0.01 < p < 0.05 (yellow), and 3) p > 0.05 (red). Only results where comparisons were conducted on the four out of six members per group are presented (for the complete set of data refer to supplementary information).

For the 0- and 180-degree groups at 5 psi BOP, all but two comparisons resulted in exceeding the threshold of statistical significance (p < 0.01, upper left and lower right quadrants in Fig 6A). On the other hand, only one pair exceeded the p < 0.05 threshold in the multiple comparison test among the eight included in the 90- and 270-degree groups (marked in yellow in the upper right and lower left quadrants in Fig 6A). A similar, but somewhat less pronounced relationships were observed in the impulse group (Fig 6B). Increasing the nominal BOP to 10 psi also increased the number of pairs, where the lowest threshold of statistical significance is observed (p < 0.001, 8 out of 16 in Fig 6C compared to 4 out of 16 in Fig 6A). The matrix of pairwise comparison for the 10 psi impulse results in the highest number of pairs where the lowest threshold was exceeded (p < 0.001, 10 out of 16, Fig 6D). When one of the incident pressure sensors is eliminated from the peak overpressure average, the threshold of statistical significance threshold is not met (i.e., in FLB and FRB for 0° and 180° orientation, and LRB and FLR for 90° and 270° orientation, respectively). However, when one of the sensors which belong to the reflected pressure groups is eliminated, it results in averages being statistically significant. (Note: The "reflected pressure" group is defined here as a sensor facing the shock wave and the one located on the back (e.g., FRONT and BACK sensors in the 0° orientation, or RIGHT and LEFT in the 90° orientation, and so on.) Interestingly, we noticed that irrespective to the orientation of the helmet and nominal incident BOP, the pairwise comparison returned no statistically significant results when the impulse was considered for the FRB-FLB and LRB-FLR pairs (see data set "Fig 6" in the S1 File).

## Field validation tests

The laboratory tests performed in the shock tube were subsequently validated in field tests using two different helmets (Ops-Core and Enhanced Combat Helmet, ECH, Fig 7A). The headforms with helmets were mounted 12 m from the charge affixed to the wall. The incident overpressure values calculated using the average of LEFT and RIGHT sensors indicate that the peak overpressure is comparable for both helmets (10.5 and 11.0 psi, for Ops-Core and ECH, respectively). Similarly, to the shock tube data, the FRONT and BACK sensors reported higher peak overpressure values compared to the sensors mounted on the side, irrespective of the helmet used. Interestingly, the reflected peak overpressure was higher for the ECH helmet compared to the Ops-Core (16.2 vs. 13.0 psi, respectively), while the peak overpressure for the BACK sensor difference was merely 1 psi (14.0 and 15.0 psi, respectively, Fig 7B).

## Discussion

The Public Law No: 116–92, FY 2020 National Defense Authorization Act, SEC. 742 direct the Department of Defense to conduct a longitudinal medical study on blast pressure exposure of members of the armed forces, and a modification of requirements for longitudinal medical study on blast pressure exposure of members of the armed forces and collection of exposure information [10]. Presently there are no recommendations on the practical implementation of these legislative obligations. The standards of the data collection and technical specifications for wearable pressure sensors don't exist. The number of sensors, their location, and post-

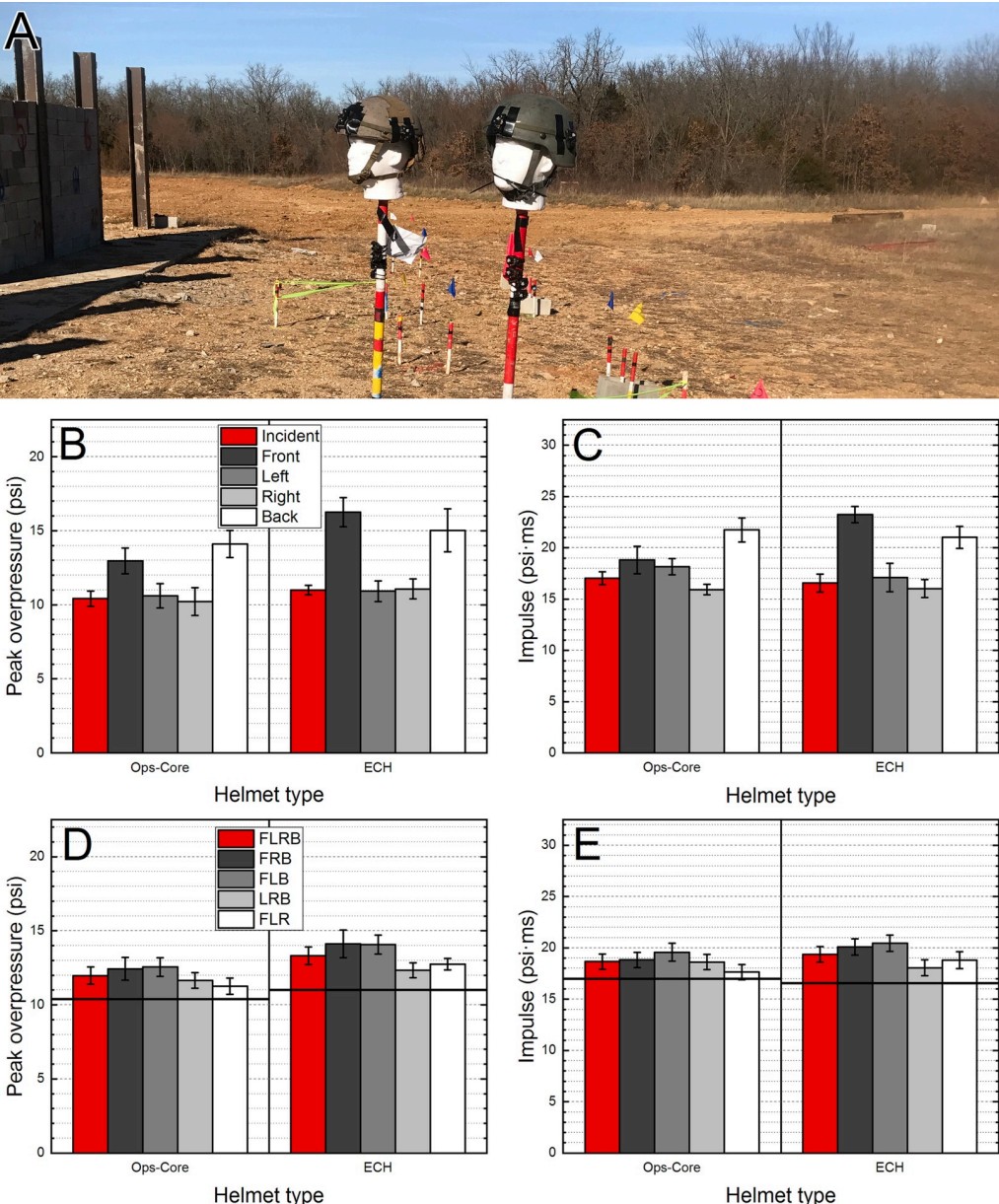

**Fig 7.** The field validation tests were performed at the heavy wall breaching exercise using two headforms mounted on range poles and retrofitted with Ops-Core and ECH (A, left and right, respectively). The same sensoring scheme as for shock tube testing was used. The quantification of the peak overpressure (B) and impulse (C) is presented, with a complementary analysis of the four and three sensor averages for peak overpressure (D) and impulse (E), respectively. For sensor notations refer to the text and Fig 5. The horizontal black line indicates the incident peak overpressure (or impulse) calculated as an average of two sensors: LEFT and RIGHT.

processing algorithms are subject of contention, and focused, systematic research is needed to establish the knowledge base necessary to formulate these standards. We focused on three important aspects: performance of currently utilized wearable sensors, their orientation, and evaluation of rudimentary post-processing algorithms. We performed this evaluation at the two pressure levels, which are within typical range likely to occur in the military training environment [2, 3].

Our research clearly demonstrates that the orientation of the blast gauge(s) will present some degree of error in the blast exposure value. The reader must also be mindful of the fact that the body positioning of the individual wearing the gauge(s) will also cause some additional degree of "error." The degree of error cannot be known or adjusted for if the following are not identified by the person interpreting the data: 1) the wearer's gauge being blocked by the blast blanket or by the individual directly in front of him; 2) the individual's head is tilted in a downward attitude; 3) the individual is positioned near a wall or other reflective surface; and 4) the individual's equipment load-out, which can also significantly impact the data.

## The fidelity of overpressure recording

We evaluated the accuracy with which the B3 sensors capture the incident and reflected pressure waveforms (Fig 1). The peak overpressure and impulse values were used as two quantitative parameters for this purpose, and the average of at least 10 overpressure waveforms recorded at nominal BOPs of 5 and 10 psi were included in the evaluation. We used the PCB sensors as a reference (Fig 2A) and found that: 1) there is good agreement between both sensor types with respect to the incident peak overpressure, but at 5 psi BOP Blast Gauges report higher values (4.96 vs. 6.15 psi, respectively, a 24% increase), 2) the peak reflected overpressure is consistently underreported by the B3 sensors, with a discrepancy in the 23–30% range depending on the nominal BOP. In Zhao's research (1999), a similar discrepancy was observed for Kulite sensors and the theoretical calculations [28]. The diminished values of the peak reflected pressure are observed consistently at both nominal shock wave intensities (5 and 10 psi). It leaves one with the conclusion that the pressure measurements of the B3 sensor gauges may provide inherently errant data when the sensor is subjected to a direct loading by a shock wave. The explanation for this discrepancy is the difference in the dynamic response of both types of pressure sensors. The typical rise times are in the 6.2–6.9 microseconds range for the 5 and 10 psi incident BOP, while the rise time is in the 1–2 microseconds for the reflected overpressure with a peak in the 15–30 psi range in our tests (Fig 1D and 1F). These values correspond to the loading rates of 0.8–1.4 psi/µs (incident) and 12–13 psi/µs (reflected). It is a tenfold difference, which possibly is more than the B3 pressure transducer can handle, judging by the rise times for these sensors (insets in Fig 1C and 1E). The sampling frequency difference between B3 and PCB sensors (200 kHz vs. 1 MHz, respectively) is not a causative. In our recent paper we demonstrated that sampling rates as low as 10 kHz are sufficient to qualitatively capture the peak pressure for shock waves with a duration of a few milliseconds [25], similar to these used in this work.

The evaluation of the impulse waveforms revealed that B3 sensors consistently overestimate these by a margin of 11–23%. The consistency of these results likely stems from the algorithm of pressure waveform integration embedded in the firmware of the B3 sensors, which uses the entire 20 ms of the recorded signal, i.e., it adds the integral of the baseline noise to the reported value. Consequently, the ratios of the reflected-to-incident peak overpressure are also underestimated for the B3 Blast Gauge (Fig 2C).

## The effect of ACH helmet orientation on the overpressure

For a flat surface facing the shock wave, the highest reflected pressure will be observed when the incident angle is 0˚ (the direction of the propagation of the shock wave is perpendicular to the surface) [19, 29, 30]. The reflected pressure is a function of the incident angle, and it will be reduced to the incident pressure at the incident angle of 90 degrees. This simplified model explains why for the ACH helmet, at the zero-degree orientation, the FRONT sensor reports peak reflected overpressure values which are higher than the LEFT and RIGHT sensors. The

incident angle for the LEFT and RIGHT sensors approaches 90˚, and these two can be used as a reasonable approximation of the incident overpressure when experimental conditions won't allow deployment of the high precision instruments (e.g., weapon training or combat deployment scenarios). In all tested helmet-headform orientations, there are always two sensors, which can be used to calculate the incident pressure substitute, i.e., LEFT and RIGHT for 0˚ and 180˚ orientations, or FRONT and BACK sensors for 90˚ and 270˚ orientations (Fig 3).

The reflected-to-incident overpressure ratio (reflection coefficient) is used to demonstrate the effect of the shock wave impacting a flat surface, and effects of the incident angle and blast wave intensity (see Fig 4-2 in FEMA-426 reference manual [30], and Fig 2-193 in DoD UFC manual [19]). This ratio (also referred to as amplification factor [21]) is a non-dimensional parameter, which is convenient for analysis of pressure distribution on the surface of the subjects with complex geometries. As expected, the analysis of the amplification factors indicates that these are a function of the shock wave intensity. For the 5 psi BOP, the average peak overpressure amplification factors are in the 1.3–1.5 range (Fig 3E), while for 10 psi BOP, these are in the 1.7–1.95 range (Fig 3F). The same is true for the impulse amplification factors: 1.15–1.3 for 5 psi BOP (Fig 4C), and 1.4–1.55 range for 10 psi BOP (Fig 4D).

The prediction of blast effects on buildings is typically conducted by employing empirical, semi-empirical and numerical methods. The general algorithm encompasses two steps: 1) the free-field (incident) blast curve is calculated, and 2) the reflected overpressure loading is applied on the building surfaces [31, 32]. Moreover, in the prediction of blast effects on structures the impulse is always incorporated in the analysis as one of the important parameters. On the other hand, the blast injury thresholds in humans were established using a maximum effective pressure, defined as "the highest of incident pressure, incident pressure plus dynamic pressure, or reflected pressure." In this way the thresholds of eardrum rupture, lung damage, and lethality are defined (see Table 1–1 on page 21 in UFC 3-340-02 [19]). In the context of occupational low-level blast monitoring, it is thus highly desirable to document incident pressure, considering it is recorded with smaller error than reflected pressure by currently available wearable sensors.

## Averaging of overpressure characteristics

In the next step, we performed an analysis of the average peak overpressure and impulse using either all four sensors or by using a combination of three sensors (Fig 5). This analysis was performed to evaluate the practical aspect of the occupational exposure measurements where the precise characteristics of the incident shock wave (peak overpressure and impulse) and its directionality might be ambivalent. The question regarding the number of sensors which are needed to calculate an accurate overpressure dosage remains open-ended, and there are currently no guidelines that define this crucial parameter. This work, is thus, the first of its kind to demonstrate quantitative relationships observed when a four-sensor setup is deployed to capture the occupational BOP exposure. It is worth mentioning at this point that the sensoring scheme for the monitoring of occupational exposure recommended by the manufacturer of the B3 Blast Gauges uses three sensors mounted on the chest, shoulder, and back. While our tests don't replicate this configuration, parallel conclusions can be drawn from our experiments, considering the three-sensor setup is approximated, in our case, by the elimination of one among the four sensors used.

The results of our study suggest that for peak overpressure the contribution of the reflected pressure elevates the average values by a margin as high as 18–22% and 45–50% (the FRB and FLB configurations in Fig 5A and 5B, respectively) at 5 and 10 psi nominal shock wave intensity compared to the calculated incident pressure values (indicated by a horizontal black line).

Similarly, for the impulse in these configurations, the averages were overestimated by 12–15% and 23% (Fig 5C and 5D). Eliminating the reflected pressure sensor from the calculations brings the calculated average numbers closer to the two-sensor incident pressure values. However, for the impulse averages, in all but one cases, the averages of three-sensors are overestimated, regardless of the helmet orientation and nominal shock wave intensity.

Fig 6 presents a heatmap of p-values calculated using the multiple pairwise comparison t-test between two classes of three-sensor averages: one class has removed FRONT or BACK sensor, while the second class members have LEFT or RIGHT sensor excluded. This heatmap demonstrates the variability of results among three-sensor averages via statistical significance thresholds. The results of our analysis indicate that averaging of the various combinations of the sensors creates a matrix of highly disperse results, and the thresholds of statistical significance depend strongly on the direction of the incident shock wave (helmet orientation) and its intensity.

For the field tests, the calculated average values for the peak overpressure are the highest for these sensor combinations where both FRONT and BACK sensors are included (FLB and FRB, Fig 7C). This is the same pattern we observed in the shock tube tests performed in the 0˚ orientation (Fig 5A and 5B). The impulse variation follows similar trends, with the peak values for the LEFT and RIGHT sensors matching closely to the incident impulse values, and increased values for the FRONT and BACK sensors (Fig 7D). The only difference from the peak overpressure quantification is the FRONT sensor value for the Ops-Core helmet, which didn't reach the threshold of statistical significance (p = 0.02). The differences in the tests aimed at evaluating the effects of the averaging of the impulse (Fig 7E) indicate that, similarly to the peak overpressure counterparts (Fig 7C), the average impulse values are overestimated in all cases. In general, these values are overestimated by a margin of one standard deviation or more. Thus, it can be concluded that at the evaluated overpressure level (10–11 psi), regardless of the helmet used in the tests, the results are consistent with the findings from the detailed shock tube evaluation.

As a point of context, the reader should understand that when reporting BOP data, with a known error of one standard deviation, what impact that will have on a unit's combat readiness. For example, if a soldier is exposed to an explosive event while wearing a "DARPA configuration" of B3 sensors, and that data is analyzed, what BOP value should be reported to the unit's medical department? If the reflected pressure is reported, which the authors have shown could be nearly twice the actual BOP value, the service member may be medically evacuated due to exceeding the permissible exposure level. Thereby, unnecessarily reducing the unit's combat effectiveness.

It is worth to underline that the sensor evaluation was performed at two pressure levels experienced in training by the military, where typical range is 1–12 psi [2]. Our findings indicate that incident pressure is superior choice for the "exclusion from duty" metric because it does not depend on the trainee and sensor orientation, and local geometry. Trends observed in the results of pressure averaging are transferable to other sensor combinations; a higher relative contribution of the reflected pressure in the average will drive the average value further away from the incident pressure. We have also identified inadequate dynamic response of wearable sensors results in reflected pressure undermeasure, that is likely to aggravate data analysis if unresolved. The limiting factor of this work is that generation 7 of the Blast Gauge is currently available. These sensors have increased onboard storage capacity and implemented wireless communication capabilities, but there is no information regarding pressure transducer changes. Moreover, the averaging analysis was done on uncorrected peak overpressure and impulse values, as these values would not be known generally without reference pressure gauges. Applying corrections would likely lead to higher divergence of results, particularly in

peak overpressure evaluation. The impulse values in both shock tube and field tests (approx. 17 psi·ms and 4 ms duration (incident)) are also on the upper end of these experienced in training. An additional study that uses short duration/lower impulse shock waves representative for other training environments would be needed to confirm our findings. Lastly, the intermediate helmet orientation angles, e.g., 45 degrees, might result in different pressure distribution patterns at the four reference points used in this work.

## Conclusions

We compared the performance of two types of sensors, the wearable B3 Blast Gauge® and the industry standard high-frequency response PCB transducers, the latter used as a reference. Results of our analysis revealed that: 1) there is a large negative error in reflected pressure for B3 sensors, but incident peak overpressure is comparable, 2) impulse values are overestimated, regardless of the test configuration (incident or reflected overpressure) for B3s.

We performed a systematic evaluation of the effects of sensors orientation with respect to the direction of the shock wave propagation on the pressure distribution on the ACH. Averaging of sensor readings (clustered as a combination of four or three sensors) leads to considerable variability of results. In general, the inclusion of the reflected pressure sensors leads to an increased peak overpressure and impulse values. This increase scales with the proportion of the reflected overpressure contribution included into the average. Finally, we noted three and four sensors averages have predominantly higher values than the incident pressure.

These findings point towards extreme caution, which should be exercised while interpreting the occupational blast exposure results from a single or a few sensor clusters regardless of their spatial distribution. Unadjusted wearable sensor-recorded values without the geometrical, training regime specific information such as orientation of the trainee with respect to the source of the blast wave; and weapon system used might lead to incorrect estimation of the individual and cumulative dosages. Considering the performance issues of the wearable sensors identified herein, the incident overpressure appears as a more desirable candidate to report the occupational exposure levels.

## Supporting information

**S1 File.**
(ZIP)

## Acknowledgments

Material has been reviewed by the Walter Reed Army Institute of Research. There is no objection to its presentation and/or publication. The opinions or assertions contained herein are the private views of the author, and are not to be construed as official, or as reflecting true views of the Department of the Army or the Department of Defense.

Help received from Drs. Walter Carr (MAJ., Retd.) and Michael Egnoto, and Bradley A. Garfield (CW5, Retd.) during the preparation of this manuscript are gratefully acknowledged.

## Author Contributions

**Conceptualization:** Maciej Skotak, Namas Chandra, Gary H. Kamimori.

**Data curation:** Anthony Misistia, Maciej Skotak.

**Formal analysis:** Anthony Misistia, Maciej Skotak, Eren Alay.

**Funding acquisition:** Gary H. Kamimori.

**Investigation:** Anthony Misistia, Arturo Cardenas, Eren Alay.

**Methodology:** Anthony Misistia, Maciej Skotak, Eren Alay, Namas Chandra, Gary H. Kamimori.

**Supervision:** Gary H. Kamimori.

**Visualization:** Maciej Skotak.

**Writing – original draft:** Anthony Misistia, Maciej Skotak, Gary H. Kamimori.

**Writing – review & editing:** Maciej Skotak, Gary H. Kamimori.

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
