## [Decision Letter · Decision Letter 0]

30 Jun 2020

PONE-D-20-17170

Sensor orientation and other factors which increase the blast overpressure reporting errors

PLOS ONE

Dear Dr. Skotak,

Thank you for submitting your manuscript to PLOS ONE. After careful consideration, we feel that it has merit but does not fully meet PLOS ONE’s publication criteria as it currently stands. Therefore, we invite you to submit a revised version of the manuscript that addresses the points raised during the review process.

We look forward to receiving your revised manuscript.

Kind regards,

Firas H Kobeissy, PhD

Academic Editor

PLOS ONE

Additional Editor Comments:

we read with interest the work of Skotak et al,

there are major concerns related to the methodology and data analysis.

The reviewers suggested some experimental modifications that could lead to better outcomes which I totally concur.

we look forward for your response.

Thank you

Journal Requirements:

4. Please include a copy of Table 1 which you refer to in your text on page 23.

Reviewers' comments:

Reviewer's Responses to Questions

**Comments to the Author**

1. Is the manuscript technically sound, and do the data support the conclusions?

Reviewer #1: Yes

Reviewer #2: Yes

2. Has the statistical analysis been performed appropriately and rigorously? 

Reviewer #1: Yes

Reviewer #2: Yes

3. Have the authors made all data underlying the findings in their manuscript fully available?

Reviewer #1: Yes

Reviewer #2: Yes

4. Is the manuscript presented in an intelligible fashion and written in standard English?

Reviewer #1: Yes

Reviewer #2: Yes

5. Review Comments to the Author

Reviewer #1: References for BoP 2 psi should be supplied; l 63 Clarify helmet locations here and in Lm113;l 138: Delete building references use biological findings where available; l 397 Is "medical evacuation necessary? What clinical criteria? This may be force depleting based on ?? P measurement

General: This paper needs a section on strengths and limitations.

Reviewer #2: The authors present a very important study for the blast community that will help researchers have a better understand blast instrumentation, specifically pressure sensors. This fundamental knowledge is a necessity for all blast experiments and it is a significant flaw in many historical studies. If the basic knowledge of sensor reporting and its data collection are not legitimate, study results cannot be comparable and lead to misunderstanding of results that effect military and civilian policies. This is very important work, however, major revision are needed for better clarity of the study and its results.

• A significant concern is regarding the different sampling rats used for the shock tube testing and the effect it could have on the results. The authors report a large difference in sampling rate between B3 and PCB sensors. If B3 is limited, the PCB sensors could have been sampled at the same lower rate to ensure differences are due to sensors and not sampling rate. This would answer many question about the data. The authors should repeat the PCB tests with the 100 kHz sampling rate. This may have significant effect on the collection of data. Is the ‘lower’ peak pressure because of the lack of capturing the actual peak? It is important to find the answer to this question.

• Understand sensor placement is critical to read the data. Figure 1 should be revised to indicate where each sensor is located so the reader can better identify which sensor is.

• Figure 1 legend needs to be revised as states panel ‘D’ twice instead of ‘E’

• How many of each sensor type used was a bit confusing? These is some detail later in the results but should be in original methods section.

• The dimensions of the fixture shown in Figure 1 should be provided in the methods. It appears that is would take up a large portion of the shock tube cross-sectional area.

• Methods are repeated and even more thoroughly explained in the results section – could be concisely stated in the methods to begin with.

• Why were such low pressures used (5 and 10 psi)? To mimic breaching exercises? This could be justified/discussed within the introduction. Currently only a sentence in the abstract.

• Duration for all test should be listed. This might help explain the impulse differences. Was one duration time used for all calculations?

• Why was a different helmet used in field-testing than in lab testing? ACH vs ECH

o Similarly, why was a different head form used?

• “The incident peak overpressure is higher at 5 psi, and matches the PCB sensors at 10 psi.” Line 193, could use some clarifying – B3 sensor is higher/lower than PCB reference sensors?

• Says at line 192 that the impulse values are overestimated, but in line 195 says they are largely underestimated for B3 sensors?

• Justification for some experimental design aspects is lacking, e.g. why did they select the averaging techniques?

• Post processing of data is not mentioned and should be described.

• Overall, there are several sentences with unnecessary commas and sentences where commas would have been very helpful to clarify the point. Grammatical proofreading would make this much easier to read.

6. PLOS authors have the option to publish the peer review history of their article (what does this mean?). If published, this will include your full peer review and any attached files.

Reviewer #1: No

Reviewer #2: **Yes: **Pamela J VandeVord

---

## [Author Response · Author response to Decision Letter 0]

10 Aug 2020

Responses are in the file "Response to reviewers" attached to this submission.

---

## [Decision Letter · Decision Letter 1]

14 Sep 2020

PONE-D-20-17170R1

Sensor orientation and other factors which increase the blast overpressure reporting errors

PLOS ONE

Dear Dr. Skotak,

Thank you for submitting your manuscript to PLOS ONE. After careful consideration, we feel that it has merit but does not fully meet PLOS ONE’s publication criteria as it currently stands. Therefore, we invite you to submit a revised version of the manuscript that addresses the points raised during the review process.

We look forward to receiving your revised manuscript.

Kind regards,

Firas H Kobeissy, PhD

Academic Editor

PLOS ONE

Additional Editor Comments (if provided):

Dear Authors,

please modify the references accordingly and work on the rephrasing of the sentences.

The major comments have been addressed.

I think addressing reviewer #2 would be adequate for the article acceptance.

FK

Reviewers' comments:

Reviewer's Responses to Questions

**Comments to the Author**

1. If the authors have adequately addressed your comments raised in a previous round of review and you feel that this manuscript is now acceptable for publication, you may indicate that here to bypass the “Comments to the Author” section, enter your conflict of interest statement in the “Confidential to Editor” section, and submit your "Accept" recommendation.

Reviewer #1: (No Response)

Reviewer #2: All comments have been addressed

2. Is the manuscript technically sound, and do the data support the conclusions?

Reviewer #1: Yes

Reviewer #2: Yes

3. Has the statistical analysis been performed appropriately and rigorously? 

Reviewer #1: Yes

Reviewer #2: Yes

4. Have the authors made all data underlying the findings in their manuscript fully available?

Reviewer #1: Yes

Reviewer #2: Yes

5. Is the manuscript presented in an intelligible fashion and written in standard English?

Reviewer #1: Yes

Reviewer #2: Yes

6. Review Comments to the Author

Reviewer #1: Questions answered but the MS can still be made more clear by clarifying comments Eg beginning with the phrase, "It is true..." What, exactly is true? Also the charge used in the open exposures, its distance, suspension needs to be better clarified. The response to the solid wall references seems inadequate. The section on limitations needs expansion particularly comparing the open field and tube.

Reviewer #2: All concerns have been addressed and the manuscript is acceptable for publication. The correction to grammar significantly improve the read.

7. PLOS authors have the option to publish the peer review history of their article (what does this mean?). If published, this will include your full peer review and any attached files.

Reviewer #1: No

Reviewer #2: **Yes: **Pamela VandeVord

---

## [Author Response · Author response to Decision Letter 1]

20 Sep 2020

Response to reviewers' comments are in the file "Response to reviewers" attached to this submission.

---

## [Editor Report · Decision Letter 2]

23 Sep 2020

Sensor orientation and other factors which increase the blast overpressure reporting errors

PONE-D-20-17170R2

Dear Dr. Skotak,

We’re pleased to inform you that your manuscript has been judged scientifically suitable for publication and will be formally accepted for publication once it meets all outstanding technical requirements.

Kind regards,

Firas H Kobeissy, PhD

Academic Editor

PLOS ONE

Additional Editor Comments (optional):

Dear Dr. Skotak

the reviews have answered the queries.

congrats for acceptance

thank you
---

## [Editor Report · Acceptance letter]

28 Sep 2020

PONE-D-20-17170R2 

Sensor orientation and other factors which increase the blast overpressure reporting errorsSensor orientation and other factors which increase the blast overpressure reporting errors 

Dear Dr. Skotak:

I'm pleased to inform you that your manuscript has been deemed suitable for publication in PLOS ONE. Congratulations! Your manuscript is now with our production department. 

Kind regards, 

on behalf of

Dr. Firas H Kobeissy 

Academic Editor

PLOS ONE